# Agronomic, Physicochemical, Aromatic and Sensory Characterization of Four Sweet Cherry Accessions of the Campania Region

**DOI:** 10.3390/plants12030610

**Published:** 2023-01-30

**Authors:** Anna Magri, Livia Malorni, Rosaria Cozzolino, Giuseppina Adiletta, Francesco Siano, Gianluca Picariello, Danilo Cice, Giuseppe Capriolo, Angelina Nunziata, Marisa Di Matteo, Milena Petriccione

**Affiliations:** 1Department of Environmental, Biological and Pharmaceutical Sciences and Technologies (DiSTABiF), University of Campania Luigi Vanvitelli, Via Vivaldi 43, 81100 Caserta, Italy; 2Council for Agricultural Research and Economics (CREA), Research Center for Olive, Fruits, and Citrus Crops, 81100 Caserta, Italy; 3Institute of Food Science, National Research Council (CNR), Via Roma 64, 83100 Avellino, Italy; 4Department of Industrial Engineering, University of Salerno, Via Giovanni Paolo II, 84084 Fisciano, Italy

**Keywords:** autochthonous germplasm, UPOV descriptors, physicochemical parameters, bioactive compounds, polyphenols, HPLC, VOCs profile, texture

## Abstract

Sweet cherries (*Prunus avium* L.) are greatly appreciated fruits worldwide due to their taste, color, nutritional value, and beneficial health effects. The characterization of autochthonous germplasm allows to identify genotypes that possess superior characteristics compared to standard cultivars. In this work, four accessions of sweet cherry from the Campania region (Limoncella, Mulegnana Riccia, Mulegnana Nera and Montenero) were investigated for their morpho-physiological, qualitative, aromatic, and sensorial traits in comparison with two standard cultivars (Ferrovia and Lapins). A high variability in the pomological traits resulted among the samples. Montenero showed comparable fruit weight and titratable acidity to Ferrovia and Lapins, respectively. The highest total soluble solid content was detected in Mulegnana Riccia. A considerable variability in the skin and pulp color of the cherries was observed, varying from yellow-red in Limoncella to a dark red color in Montenero. Mulegnana Nera showed the highest content of polyphenols, flavonoids, anthocyanins, and ascorbic acid compared to the standard cultivars. Volatile organic compounds profile analysis identified 34 volatile compounds, 12 of which were observed at different concentrations in all the sweet cherry genotypes while the others were genotype-dependent. Conservation and cultivation of autochthonous accessions with suitable nutritional and morpho-physiologic characteristics promotes our agrobiodiversity knowledge and allows to better plan future breeding programs.

## 1. Introduction

Sweet cherry (*Prunus avium* L.) belongs to the Rosaceae family, Prunoideae subfamily, and *Prunus* genus. These fruits originated around the Caspian and Black seas, nowadays they are geographically distributed all around the world. In 2020, the annual world production was around 2.5 million tons, with the largest producers being Turkey, the United States and Chile. Italy was the sixth largest producer in the world and the largest in Europe [1].

Sweet cherry is a non-climacteric fruit, most appreciated by consumers for its sweetness, juiciness, nutritional value, and beneficial health properties [2]. Morphological and physicochemical features as well as fruit weight, skin color and firmness influenced consumer acceptance [3].

These fruits contain some phenolic compounds including phenolic acids, flavonoids, procyanidins and anthocyanins [4]. Among the anthocyanins, cyanidin-3-*O*-glucoside, cyanidin-3-*O*-rutinoside are the most abundant [4,5]. It has been demonstrated that the color of the cherry fruits depend on co-pigmentation of anthocyanins and phenolic acids [6]. Neochlorogenic acid, p-coumaroylquinic acid and chlorogenic acid are the most identified [7]. These fruits have a low glycemic index, and are a good source of vitamin C [8]. Thanks to these properties, sweet cherries are considered a functional food as they reduce the levels of fat, particularly saturated fat, the risk of cancer, pain from arthritis and inflammation and offer protection against neurodegenerative diseases [9]. The quality of cherry fruits is directly correlated with genetic variability, training systems, agronomic and post-harvest management [10,11].

Over the centuries, farmers have selected and cultivated certain varieties due to their adaptability to territories with geomorphological and environmental features [12]. Nowadays, these cultivars, also called “local varieties”, represents a significant component of agrobiodiversity and their genetic variability can be used to create new ones or even as a source of genes for improving the already existing ones so to better guarantee performances and sustainability of the production systems. New cherry cultivars can be the result of a cross between two cultivars and the subsequent selection of new cultivar candidates among the offspring. It is worth pointing out that unexpected variations in quality in the breeding progenies may happen due to hidden genetic variations in the parent cultivars [13]. Accession selection for breeding programs is important to create new genotypes with good bio-agronomic and chemical characteristics of the fruits [14]. In recent years, breeding programs have extended the cherry harvesting season that it was previously very short, with fruit being harvested between May and mid-June in the northern hemisphere and between mid-October and January in the southern hemisphere [15,16]. In Italy a wide array of cultivars/accessions are present but the most widely grown cherries are the commercial cultivars, Burlat, Ferrovia, Malizia, and Durone di Vignola [17].

The high-quality characterization of autochthonous germplasm in terms of fruit size, firmness, highly bioactive compound content allows the selection of useful genotypes for future breeding programs. However, in addition to these properties, good sensory characteristics, such as taste and flavor, have generally been overlooked. The pedological and climatic conditions of the Campania region (Italy) have contributed to the spread of several local sweet cherry cultivars and the conservation of agrobiodiversity [3,18,19]. 

Muccillo and collaborators [12] reported that Limoncella, Montenero, Mulegnana Riccia and Mulegnana Nera according to genetic, environmental, and geo-pedological parameters were featured in three different clusters (Montenero cluster 1, Limoncella cluster 2, Mulegnana Nera and Mulegnana Riccia cluster 3). In addition, Mulegnana Riccia and Nera were found to be genetically comparable. Guarino et al. [19] reported the molecular characterization of fifty-three ancient Campania genotypes of sweet cherries compared to seven standard cultivars. This study showed that Montenero, on a genetic basis, was located in cluster 3, Mulegnana Riccia into cluster 4 and Limoncella in cluster 6. 

This study aimed to perform an in-depth characterization of four sweet cherry genotypes based on their morpho-physiological, physicochemical, and sensory features. The total content of phenols, flavonoids, condensed tannins, anthocyanins, and ascorbic acid as well as the anti-radical capacity, have been evaluated in sweet cherry genotypes. Furthermore, the aromatic and phenolic profiles have been recorded by solid-phase microextraction followed by gas chromatography–mass spectrometry (SPME/GC-MS) and reverse-phase high-performance liquid chromatographic-diode array detector (RP-HPLC-DAD), respectively. 

## 2. Results and Discussions 

### 2.1. Morpho-Physiological Descriptions

The Union for the Protection of New Varieties of Plants (UPOV) guidelines for sweet cherries with forty-two morphological trait descriptors have been used to carry out tests for distinctness, homogeneity, and stability (TG/35/7-UPOV 2006) among different cultivars. The results of the pomological and physicochemical analyses and UPOV descriptors of the four sweet cherry accessions revealed differences and similarities among them and resulted in an accurate agronomic description of each accession (Appendix A). 

The Mulegnana Riccia tree has medium vigor, spreading habit, and medium productivity. The leaves have a lanceolate-elliptical shape, with toothed margins and a long petiole. Flowering is medium. The Mulegnana Riccia tree produces a medium-sized cherry, cordiform in the longitudinal section, spheroidal-appressed in the cross section, and has a medium, superficial stylar scar. The pedicel cavity is wide and deep. The cherry skin is dark red, with speckled blush. The cherry has a superficial suture line and medium red flesh. This fruit has medium texture, good taste quality, is sweet with medium acidity and slightly sour, is very succulent, and has a coarse texture. The fruit is adherent to the stone and produces red juice. The stalk has a medium length and detaches easily from the branch. The stone is small and has a broad elliptic shape with a medium dorsal ridge. The ripening period is intermediate (early June) (Figure 1a).

The Mulegnana Nera tree has medium vigor, spreading habit, and medium productivity. The leaves are medium-sized and have an elliptical-enlarged shape, with toothed margins and a long petiole. Flowering is medium. Cherries of the Mulegnana Nera tree are medium-sized with a kidney shape in the longitudinal section and spheroidal-depressed in the cross section. The fruits have a small, deep stylar scar. The pedicel cavity is small and medium in depth. The skin of the cherries is dark red, with uniform blush and a superficial suture line. The flesh is dark red in color. These fruits have medium texture, good taste quality, are medium sweet and medium succulent with medium-coarse texture. The fruit is semi-adherent to the stone. The juice is red in color. The stalk is long and easily detaches from the tree. The stone is medium-sized and has a broad elliptic shape with a shallow dorsal ridge. The ripening period is intermediate (early June) (Figure 1b).

The Montenero tree has medium vigor, spreading habit and high productivity. The leaves are large and have an elliptical-enlarged shape, with toothed margins and a long petiole. Flowering is medium. The Montenero cherry is large, kidney-shaped in the longitudinal section, and spheroidal in the cross section with a medium deep stylar scar. The pedicel cavity is wide and medium in depth. It has a red skin color, with uniform blush and a superficial suture line. The flesh is red in color. This cherry has high texture, good flavor quality, with low sweetness tending to sour, medium juicy and a medium-coarse texture. The flesh is semi-adherent to the stone. The juice is red in color. The stalk is medium with easy detachment from the branch. The stone has a medium-large size with a broad elliptical shape and a very prominent dorsal ridge. The ripening period is early (mid-May) (Figure 1c).

The Limoncella tree has medium vigor, semi-upright habit and medium-low productivity. The leaves are medium and have an elliptical shape, with toothed margins and a long petiole. Flowering is medium. The Limoncella cherry is medium-sized, has a cordiform shape in the longitudinal section and a spheroidal-depressed shape in the transverse section. The stylar scar is small and deep, while the pedicel cavity is small and medium in depth. The skin is yellow-red, with shaded blush, while the flesh is yellow. The suture line is weakly conspicuous. The fruit texture is medium, with medium flavor quality, sweet taste and very juicy. The flesh is semi-adherent to the stone. The juice is light yellow. The stalk is long in length with easy detachment from the branch. The stone is medium in size with a broad elliptical shape with a medium dorsal ridge. The ripening period is early (mid-May) (Figure 1d).

This is the first study that described all the phenological stages in these sweet cherry genotypes. Nowadays, in literature has only reported the biometric characterization of several autochthonous sweet cherry cultivars of the Campania region using UPOV descriptors [3].

### 2.2. Morpho-Physiological, Physicochemical and Sensorial Analyses

The results of morpho-physiological and physicochemical trait characterization are reported in Table 1. Cherry varieties’ fruit weight is influenced by crop load and fruit maturity stage [20,21].

Significant differences in agronomic and physicochemical traits are due to genetic diversity among sweet cherry accessions. The average fruit weight changed significantly among the accessions, with a minimum of 4.85 ± 0.37 g in Mulegnana Nera and a maximum of 8.61 ± 0.36 g in Lapins. The Montenero accession was the only one that came close in fruit weight to the standard cultivar Ferrovia.

Furthermore, a wide variation among the accessions was registered, the fruit’s average height and width ranged from 18.20 to 23.93 mm and from 20.42 to 26.90 mm, respectively. The Mulegnana Nera and Mulegnana Riccia cherries did not showed statistically significant changes in their fruit’s width or length, while among the other accessions statistically significant differences were recorded. Montenero showed the highest fruit thickness, followed by Limoncella, Lapins and Mulegnana Nera; while Mulegnana Riccia and Ferrovia showed the lowest ones. Currently, in the scientific literature few characterization studies of Campania accessions of sweet cherries have been published. Di Matteo et al. [3] characterized autochthonous Campania cherries grown in Salerno, highlighting that the morphometric and quality parameters depend on the soil and pedoclimatic conditions. Our results agree with Di Matteo et al. [3] that reported a fruit weight of out 9, 6 and 7 g for Montenero, Mulegnana Nera and Mulegnana Riccia, respectively. Taiti et al. [22] described three Tuscan sweet cherry accessions by comparing them to the standard cultivar Ferrovia. Their results agree with those reported in this paper, in particular Ferrovia was found to have a weight of about 7 g, a height of about 21 mm, a width of about 24 mm, and a thickness of about 21 mm. Moreover, they reported that the cultivars Di Giardino and Di Nello had a fruit weight like our Mulegnana Nera and Mulegnana Riccia [3]. Ballistreri et al. [4] evaluated 24 sweet cherries, many of which are typical of the Sicily region, whose weight ranged from about 4 (Ducignola Nera) to 13 g (Early Star). According to Hayaloglu and Demir [23], the weights of 12 varieties of sweet cherry growing in Turkey significantly varied between 6 and 8 g. However, a weight of 8.8 to 14.5 g was established for the genotypes grown in Canada, which is higher than our findings [24].

Important physicochemical parameters, such as total soluble solids content (TSS), pH and TA, influence consumer preference for fruit quality [25,26]. Total soluble solid estimates the level of dissolved sugars (sucrose, glucose and fructose) which increase in response to ripening [25]. In our study, the lowest TSS content was recorded in Montenero, while the highest was in Mulegnana Riccia followed by Lapins and Ferrovia. An intermediate content of TSS was recorded in Limoncella and Mulegnana Nera. Our results agree with Di Matteo et al. [19] which reported a value of 17.20 °Brix for Montenero, 21.10 °Brix for Mulegnana Nera and 18.60 °Brix for Mulegnana Riccia. Ballistreri and coworkers [4] reported that total soluble solids content of 24 sweet cherries cultivars varied from 13.53 °Brix in Sweet Early to 22.73 °Brix in Black Star. Similar findings for the Spanish Picota and Sweetheart cultivars (13.97–23.20 °Brix) have also been reported by Serradilla and others [27]. In a research by Hayaloglu and Demir [23], TSS (°Brix) values ranged between 13.26 (Summit) and 19.55 (Bing). Numerous cultivars’ TSS varied between 14 and 16 °Brix [28]. For 12 cultivars, Girard and Kopp [24] recorded TSS values ranging from 13.5 to 25.5 °Brix. Values from 12.3 to 23.5 °Brix were reported by Gonzalez-Gomez [29] and colleagues for six different sweet cherry varieties. Papapetros et al. [30] reported TSS values ranging from 14.7 to 12.8 °Brix in six sweet cherry samples collected from Greece. Likewise, Wen et al. [31] showed TSS values ranging from 17.77 to 19.97 °Brix for Chinese sweet cherries Hongdeng, Hongyan and Rainier. Thus, the total soluble solids were higher, compared to the two standard cultivars, in Mulegnana Riccia, this could be a plus point for the inclusion of this accession in future breeding programs.

Depending on the type of sweet cherry, the pH levels varied greatly. The lowest pH was observed for Lapins, followed by Montenero and Mulegnana Riccia, whereas the highest pH was observed for Ferrovia and Mulegnana Nera. For 12 cultivars, the lowest and maximum pH measurements varied from 3.56 in Sweetheart to 3.80 in 0-900 Ziraat [23]. Our results are comparable to those of Girard and Koop [24] and lower than those of Gonzalez-Gomez et al. [29] and Serradilla et al. [24]. Wen and coworkers [28] reported pH values between 3.66 and 4.06 for Chinese sweet cherries Hongdeng, Hongyan and Rainier’.

TA estimates the organic acid content of sweet cherry fruits ranging from 10.17 g citric acid L^−1^ for Ferrovia to 12.35 g citric acid L^−1^ for Lapins. Significant differences (*p* < 0.05) were observed between all the accessions examined (Table 1). Although all the accessions considered showed a higher TA value than the cultivar ‘Ferrovia,’ they may be more appreciated by consumers than the standard cultivar Lapins.

The key criteria used to evaluate sweet cherries’ appealingness are fruit size and color [32]. The cherry’s skin color is thought to be the most significant indicator of cherry quality and maturity [33]. The skin and pulp’s color parameters are reported in Figure 2. The skin lightness (L*) values ranged from 61.40 ± 8.72 in Limoncella to 11.41 ± 0.14 in Mulegnana Riccia. The pulp L* varied from 91.48 ± 3.37 in Limoncella to 22.33 ± 1.43 in Mulegnana Riccia, this was due to the different pulp colors which were yellow for Limoncella and different shades of red in the other accessions.

Young’s module is a key parameter to describe the texture in a food sample [34,35,36]. In our study, Limoncella fruits showed the highest Young’s modulus and boundary load values (Figure 3a,b). Young’s modulus allows us to understand the stiffness of the fruit [34]. In our study, Young’s modulus was higher in Limoncella (0.70 ± 0.14 N mm^−^²) and lower in Montenero (0.29 ± 0.16 N mm^−^²) (Figure 3a). Boundary load was higher in Limoncella (30.24 ± 6.02 N) and lower in Lapins (11.02 ± 1.46 N) (Figure 3b). Higher Young’s modulus values are correlated with crack resistance and cuticular membrane thickness as suggested by Matas et al. [37] in cherry tomato fruit.

A skilled panel of 12 judges, with ages ranging from 25 to 50, assessed the sweet cherries’ sensory profile (Appendix A). A trained panel of tasters’ sensory analysis (Appendix A) revealed few significant differences between cultivars. Only three of the twelve characteristics, astringency, consistency, and appearance had noticeable differences, according to the tasters. The panel determined that Ferrovia had a higher value for appearance, Lapins was more astringent in comparison to the other sweet cherries, and Limoncella has a good consistency (Appendix A). 

### 2.3. Bioactive Compounds 

Cherry fruit quality is recognized to be primarily influenced by genotype and orchard management, which is expressed as the concentration of nutritive and bioactive chemicals [22]. Indeed, the variability in the secondary metabolites may be also explained by climatic factors, soil composition, and berry harvesting [38]. 

Polyphenols are a group of phytochemicals found in fruits that have antioxidant and free radical-scavenging properties. In addition to their organoleptic properties, these compounds are also responsible for the red color and astringency of sweet cherries [39]. The greatest levels of polyphenol were registered in Mulegnana Nera and Ferrovia with values of 435.91 ± 6.68 mg GAE 100 g^−1^ FW and 392.57 ± 16.05 mg GAE 100 g^−1^ FW, respectively (Figure 4a). Lapins, Limoncella and Mulegnana Riccia showed a comparable phenol content. The Mulegnana Nera accession showed a higher polyphenol content than both the standard cultivars, while Montenero had a higher content than Lapins, but lower than Ferrovia. For cultivars Napoleona Grappolo and Maredda, the total phenol levels varied from 84.96 to 162.21 mg GAE 100 g^−1^ FW [4]. In our study, the average total polyphenol content (316.66 mg GAE 100 g^−1^ FW) was higher than that of Kim et al. [40] and Gonçalves et al. [20] who reported mean values of 110 and 148 mg GAE 100 g^−1^ FW, respectively.

Anthocyanins gives many fruits, notably sweet cherries, their appealing red and violet-blue hues. They also have a significant amount of antioxidant activity [41]. Total anthocyanin concentration varied significantly (*p* < 0.05), from 11.78 ± 0.13 mg C3G 100 g^−1^ FW in Limoncella to 264.72 ± 1.44 mg C3G 100 g^−1^ FW in Mulegnana Nera. The genotype Mulegnana Nera exhibited the highest anthocyanin content, increased 2-fold compared to the cultivar Lapins and 4-fold compared to Ferrovia. Montenero possessed almost twice the anthocyanin content of Ferrovia (Figure 4b). According to Ballestreri et al. [4] the total anthocyanin content varied between 6.21 mg CGE 100 g^−1^ FW of the cultivar Gabbaladri and 94.20 mg CGE 100 g^−1^ FW of the cultivar Maredda. The levels of total anthocyanins found in this study was comparable to those found in previous studies [20,40].

Fruits also contain a lot of flavonoids, which serve many purposes for the plants. They are fundamental pigments for creating hues that draw pollinating insects. In higher order plants, flavonoids are also necessary for chemical communication, UV filtration, nitrogen fixation, and cell cycle inhibition [42]. The investigated genotypes Ferrovia, Lapins and Limoncella displayed comparable total flavonoid contents (about 80 mg CE 100 g^−1^ FW), which were significantly lower (*p* < 0.05) than Montenero and Mulegnana Nera (208.82 ± 0.26 and 147.43 ± 4.03 mg CE 100 g^−1^ FW, respectively); the genotype with the lowest concentration was Mulegnana Riccia (Figure 4c). Even in the case of flavonoids, Mulegnana Nera and Montenero were better organoleptically than Ferrovia and Lapins. Szpadzik et al. [12] showed that the total flavonoids content ranged between above 40 mg 100 g^−1^ FW in Jacinta and 15 mg 100 g^−1^ FW in Tamara and Kasandra cultivars. 

Ascorbic acid may be considered as the main antioxidant and component of redox signaling. It has an impact on fruit development and post-harvest storage, as well as fruit ripening and stress resistance [43]. Among the accessions considered in this study, Limoncella and Mulegnana Nera possessed the highest ascorbic acid content, followed by Montenero and Ferrovia. The lowest ascorbic acid contents were recorded in Mulegnana Riccia and Lapins (Figure 4d). Consuming ascorbic acid is crucial for human health. The accessions Mulegnana Nera, Montenero, and Limoncella displayed a content that was much higher than Lapins and similar to Ferrovia.

According to the bioactive compound content, Mulegnana Nera appeared to have greater antioxidant activity than the other genotypes examined (9.93 ± 0.12 µmol TE g^−1^ FW), followed by Ferrovia and Montenero (7.44 ± 0.06 and 7.21 ± 0.09 µmol TE g^−1^ FW, respectively). The genotypes Limoncella and Lapins showed similar activity with values of 6.15 ± 0.04 and 6.29 ± 0.03 µmol TE g^−1^ FW, respectively, followed by Mulegnana Riccia with 6.62 ± 0.07 µmol TE g^−1^ FW (Figure 4e). Szpadzik et al. [11] reported that the antioxidant activity varied between above 1 µmol TE 100 g^−1^ FW in Jacinta and below 0.5 µmol TE 100 g^−1^ FW in Helga and Kasandra cultivars.

Proanthocyanins, also known as condensed tannins, are a class of polymeric flavan-3-ols found in the fruits, vegetables, nuts, seeds, and flowers of many plants. They give wine, fruit juices, and teas their distinct flavors and astringency [44]. Regarding proanthocyanin concentration, the investigated sweet cherries showed significant variations. The highest mean was recorded for Lapins, which was 239.36 ± 1.74 mg CE 100 g^−1^ FW, followed by Mulegnana Nera, 224.19 ± 6.71 mg CE 100 g^−1^ FW, Montenero, 216.87 ± 8.13 mg CE 100 g^−1^ FW, and Ferrovia, 208.85 ± 0.52 mg CE 100 g^−1^ FW. The two accessions with the lowest content were Limoncella (124.66 ± 0.64 mg CE 100 g^−1^ FW) and Mulegnana Riccia (139.83 ± 4.75 mg CE 100 g^−1^ FW) (Figure 4f). Mulegnana Nera and Montenero showed a TCT content similar to the standard cultivars. Hu and collaborators [45] reported that the total tannin content ranged from about 0.17 mg CE g^−1^ in Lapins to 0.03 mg CE g^−1^ in the Bing cultivar. Prvulović et al. reported that the TCT value of sweet cherries ranged from 32 to 75 mg GAE 100 g^−1^ [46]. The differences between our results and Hu’s and Prvulović’s results are probably due to the different methods used to detect and express the tannin content [45,46].

Although the concentrations of secondary metabolites are cultivar-dependent as well as related to environmental stress, Mulegnana Nera appeared to have a higher amount of antioxidant compounds, so this could be a determining factor in increasing the cultivation of this accession and its inclusion in breeding programs.

### 2.4. Volatile Organic Compound Profile Analysis

Overall, 34 volatile compounds were detected by the HS-SPME/GC-MS analysis of the six sweet cherry genotypes, including aldehydes (8), alcohols (14), terpenes (7), esters (3), and others (2), as shown in Appendix A, which also lists the VOC abbreviation codes, the experimental and literature Kovats index and the identification methods. The effects of genotypes on the observed volatiles and the results are reported in Table 2. Moreover, Figure 5 indicates the percentage of all VOC classes identified in the sweet cherries, and it shows that aldehydes, followed by alcohols, were clearly the most abundant VOCs detected in all the samples.

Among the detected volatiles, 12 VOCs were observed at different concentrations in all the studied sweet cherry cultivars, including hexanal (Ald1), *cis*-3-hexenal (Ald2), 2-hexenal (Ald3), benzaldehyde (Ald7), 3-methyl-2-butenol (Al4), 1-hexanol (Al5), *trans*-3-hexenol (Al6), 2-hexenol (Al8), benzyl alcohol (Al12), phenyl alcohol (Al13), linalool (T4) and geraniol (T7). In particular, Ald1, Ald3 Al17, Al8, have been previously described among the predominate flavor contributors to the sweet cherries’ aroma [47].

Table 2 indicates that some of the VOCs were detected only in one cultivar, next to each value is a letter identifying significance for *p* < 0.05, while on the side, in the form of asterisks, is the significance for *p* < 0.0001 and 0.001. These volatiles may thus be used as putative cultivar-specific aroma compounds. In this regard, decanal (Ald6) was only found in Lapins sweet cherry cultivars, β-ocimene (T2) was exclusively present in Limoncella samples, *trans*-2-hexenyl butyrate (E3) was only detected in the VOC profile of Montenero and 3-methyl-3-buten-1-ol (AL3) was only observed in Mulegnana Nera.

The volatile profile of several sweet cherries reported in literature show commonality within the cultivars [23,48,49,50,51], but the distribution and content of these metabolites is highly cultivar dependent and so is the final perceived aroma [47].

Carbonyl compounds and alcohols showed the most abundant signals in the aroma profile of all fruit samples, as expected [47], and, although, Ferrovia, Lapins and Mulegnana Nera showed a similar percentage of aldehydes and alcohols with respect to the total VOC amount, alcohols were the predominant component in all cultivars, ranging from 72.7 to 49.3% of the total volatiles in Limoncella and Mulegnana Nera (Table 2). In particular, 2-hexanol (Al8) was the most abundant compound in all samples, not only among the alcohols, but also concerning all detected VOCs, with a percentage ranging from 50.0% in Montenero to 47.6% in Limoncella of the total volatiles. Serradilla et al. identified Al8 as the main alcohol present in two Spanish sweet cherries cultivars, namely Picato type and Sweetheart [27]. Al8, a product of the LOX pathway, is described by having fresh green, grassy and leafy notes, and, as mentioned above, together with hexanal and 2-hexenal, are the most relevant volatiles in the flavor of sweet cherries [47,51].

Aldehydes were the main VOCs in Ferrovia and Mulegnana Nera samples, accounting for 49.7 and 48.3% of the total volatile compounds, respectively. Among the aldehydes, 2-hexenal (Ald3) was the principal constituent in all studied cultivars (Table 2). According to Vavoura et al., Ald3 showed higher levels in Ferrovia (36.5%) compared to Lapins (29.1%) with respect to the total volatiles (Table 2) [49]. The second most abundant aldehyde was hexanal (Ald1), followed by benzaldehydes (Ald8) (Table 2). Ald3 along with Ald1, indicated as “green leaf volatiles”, are responsible for the herbaceous odor in fruit and present a very low perception threshold [49]. Benzaldehyde (Ald8) is produced by the enzymatic hydrolysis of amygdalin [51]. The relative content of Ald8 was observed to increase with fruit development and is reported among the principal contributors to the distinctive aroma of sweet cherries [48].

Ester compounds, commonly responsible of pleasant floral and fruity flavors in fruit, were completely absent in Ferrovia samples and detected in very small percentages (between 0.2 and 1.3% of the total volatiles in Mulegnana Nera and Montenero) with respect to the total VOCs in the other five sweet cherry cultivars, in line with some previous reports on other cultivars [47,49].

In general, the biosynthesis of esters is associated with a decrease in the levels of corresponding aldehydes, and in the final step of this process where acyl transferases are involved. The activity and expression of these enzymes depend on the maturity stage of the fruit; thus, ripening changes may be also correlated with the perception of the herbaceous notes [47,52,53]. In the present study, only three esters were observed (E1–E3) and, among them, hexyl acetate (E1) and *trans*-2-hexenyl acetate (E2) were detected at concentrations lower than the respective aldehydes Ald1 and Ald2 in all cultivars. Studies on food aroma have demonstrated that odor perception is ascribed to the occurrence of synergistic or suppression phenomena based on the interaction among the flavor components [41]. It has been reported that the herbaceous notes in sweet cherries are related to the presence of a higher amount of C6 aldehydes and a lower content of esters [47], and that some acetate ester compounds, including E1, are negatively correlated with the consumers’ preference [52]. These findings suggest that all the characterized sweet cherry cultivars presented a green and herbaceous aroma even if harvested under different ripening conditions. Further experiments conducted using a sensory panel of trained judges should be carried out to confirm this hypothesis.

Consistent with several studies, terpenes (T1–T7) were found in very small amounts in each cherry cultivar (Table 2) [23,30,47]. Limonene (T1) was identified at very low levels in all cultivars, except Montenero. Besides being previously detected in Ferrovia and Lapins [43], this terpene, along with T4, is commonly reported in the VOC pattern of different sweet cherry cultivars [23,30,47,48,54]. Finally, geraniol (T7), described by a rose odor and detected in all cultivars (Table 2), was previously observed in some sweet cherry cultivars harvested in different regions of Turkey [21].

The differences in the qualitative and semi-quantitative content of VOCs among the sweet cherry cultivars has been shown to be particularly valuable for the discrimination of fruit origin (genetic or geographic) [55]. For this reason, exploratory PCA data experiments were carried out to evaluate the efficacy of VOC patterns in detecting variations in the volatile content based on genotype and to identify possible volatile markers for cultivar distinction. 

The dataset was composed of 18 observations (six biological samples as technical triplicates) and 34 VOCs and the biplot achieved by modelling the HS-SPME/GC-MS semi-quantitative data (% RPA) by PCA is shown in Figure 6. The two components PC1 and PC2 accounted for 32.07% and 22.56% of the variation in the dataset, respectively. Specifically, Limoncella was characterized by both positive PC1 and PC2, while Lapins and Mulegnana Riccia displayed negative PC1 and PC2. Moreover, Mulegnana Nera and Ferrovia presented negative PC1 and positive PC2 compared to Montenero that displayed positive PC1 and negative PC2 (Figure 6). From the biplot, it can be deduced which vectors contributed to the position of the score plots of the accessions and cultivars in the graph. In particular, the Al3 vector determined the placement of the Mulegnana Riccia score plot because it is a cultivar-dependent volatile organic compound, i.e., only present in this accessions. The vectors related to the aldehydes *cis*-3-hexenal (Ald2), benzaldehyde (Ald7), and dodecanal (Ald8) and the terpenes limonene (T1) and linalool (T4) were most closely related to Mulegnana Nera. *Trans*-2-hexenyl butyrate (E3) is also a cultivar-dependent volatile organic compound, and together with the Al10 vector, oriented the position in the graph of the Montenero accession. The position of Limoncella was determined by the vector relating to VOCs Al5, Al8, T5, T6, T2, and Al13, which, as shown in the table below, are those most prominent in the accession. Decanal (Ald6), 1-octen-3-ol (Al9), and phenol (Al14) are cultivar-dependent VOCs, and they determined the position in the biplot of the Lapins cultivar. The volatile organic compounds most commonly present in the cultivar Ferrovia were hexanal (Ald1), 2-hexenal (Ald3) and geraniol (T7), the vectors of which determined the score plot position in the PCA.

The semi-quantitative variability among the volatile profiles induced by individual cultivars was visualized by the heatmap reported in Figure 7, which allowed the characterization of each cultivar in terms of its VOCs. Owing to the cultivar effect on the fruit volatile composition, different VOCs correlated with specific sweet cherry cultivars. In detail, all three detected esters (E1–E3) were positively associated with Montenero, which was the only cultivar that displayed all three metabolites in the volatile profile (Table 2; Figure 7). Montenero was also directly related to 3-methyl-2-buten-1-ol (Al4) and 1-hexanol (Al5), which have been reported to significantly increase during sweet cherry ripening [27,51]. Similarly, Mulegnana Riccia was positively correlated, among others, to E1, E2 and Al4, all of which are described to increase during ripening, and indirectly linked to 3-methylbutanoic acid (O2), which being an acid, tends to disappear along with the maturation degree [51]. These findings suggest that both Montenero and Mulegnana Riccia have reached a good state of ripening in agreement with °Brix (Table 1).

On the other hand, Ferrovia, being directly associated with the C6 aldehydes hexanal (Al1) and 2-hexenal (Ald3) and to the C6 alcohol 2-hexenol (Al8) and inversely related to Al4 and E2 (Figure 7), seems to be dominated by green and herbal notes.

Limoncella, although situated in the same quadrant of Montenero, is positively associated with four alcohols, including 1-hexanol (Al5), *trans*-3-hexenol (Al6), benzyl alcohol (Al12) and phenylethyl alcohol (Al13), three terpenes, as β-ocimene (T2), α-terpineol (T5) and myrtenol (T6) and the acid O2 and negatively correlated with 1-octanol (Al11). Terpenes, responsible for the typical citrus, fruity and floral sensory properties of fresh fruit, could impart pleasant sensory flavors to this cultivar [56].

Al11, 1-octen-3-ol (Al9), phenol (Al14), decanal (Ald6), and o-cymene (T3) are directly correlated with Lapins, which is negatively associated with two C6 alcohols 1-hexanol (Al5) and 2-hexenol (Al8) described as fresh and green (Figure 7).

Finally, Mulegnana Nera is positively correlated with *cis*-3-hexenal (Ald2), *cis-*3-hexenol (Al7), and 6-methyl-5-hepten-2-one (O1) all with a green flavor, with benzaldehyde (Ald7), described as sweet almond and cherry, and with linalool (T4) with citrus notes.

Clusters are also shown in Figure 7. The top cluster establishes the similarity, based on the VOC profiles, between the accessions and cultivars. In particular, the accessions Montenero and Limoncella stand out from the others, while there was a close similarity between the standard cultivars Lapins and Ferrovia and the accessions Mulegnana Nera and Mulegnana Riccia. The side cluster separated the volatile organic compounds into two significant groups, the first of which contained Al10, E3, T2, Al12, Al13, T5, Ald2, T4, E1, E2, Al4, Al5, Al6, and T6, and the other contained the remaining VOCs.

### 2.5. Semi-Quantitative Determination of Polyphenols

Typical RP-HPLC chromatograms of cherry phenolic compounds recorded at 340 and 520 nm are shown in Figure 8. 

These wavelengths were selected to monitor hydroxycinnamic acid derivatives and anthocyanins. Labelled HPLC peaks were assigned based on retention time order, UV–vis spectra and previous characterizations [57] and were semi-quantified as summarized in Table 3. The assignments were confirmed by HPLC coupled to high-resolution tandem mass spectrometry (MS/MS). Chlorogenic acid derivatives and the structurally correlated 4-p-coumaroylquinic acid were the predominant colorless phenolics of sweet cherries, which also contained significant amounts of quercetin derivatives, such as rutin. Colorless phenolics, especially neochlorogenic acid, varied within relatively ample ranges among the different cultivars. Similarly, anthocyanins were characterized by a wide qualitative and quantitative variability, which also reflects the color of the ripe fruits. However, the dominant anthocyanin was cyanidin 3-*O*-rutinoside in all the cultivars analyzed, while delphinidin 3-*O*-rutinoside was detected only in trace amounts with slightly variable intensity. Individual polyphenol compounds were substantially in the ranges generally reported for sweet cherries [3]. The anthocyanin content of Mulegnana Nera was particularly high, in line with the deep purple color of these cherries. Mulegnana Nera was also the richest in hydroxycinnamic acid derivatives, whereas the ranks relevant to the content of anthocyanins and colorless phenolics were quite different (Figure 4b, Table 3). The level of anthocyanins has agronomic and commercial importance, considering that these vacuolar pigments exert protective effects for the fruits against the most energetic components of sun light in the pre- and post-harvest stages, and the color is among the primary factors influencing the consumers’ choice. The extracts of Mulegnana Nera cherries also exhibited the highest antioxidant activity, reflecting the high content of phenolic compounds (Figure 4e), although ascorbic acid also contributes significantly to the antioxidant potential. 

## 3. Materials and Methods

### 3.1. Plant Materials

Six sweet cherry genotypes were selected for this study: four local accessions of the Campania region (Italy) (Mulegnana Nera, Mulegnana Riccia, Limoncella, and Montenero), one typical of the Puglia region (Ferrovia) and one of the Canada state (Lapins) that represent the standard accession. The Campania genotypes were collected in an experimental orchard located in Pignataro Maggiore (Caserta. Italy) at the CREA Research Centre for Olive, Fruit and Citrus Crops (41°04′ N. 14°19′ E with an altitude of 61 m above sea). Plants were observed in situ and agronomic traits were observed in different seasons according to descriptors listed in the TG/35/7 test guidelines released by the International Union for the Protection of New Varieties of Plants (UPOV) in 2006. Flowers, leaves, and fruits were transported to the laboratory for further assessment. Fruits were harvested from three trees from May to June at the commercial ripening stage, 89 of BBCH scale as suggested by Fadón et al. [58] and they were transported to the laboratory, screened for uniformity, appearance, and the absence of physical defects or decay for further analyses. Trees were purchased and planted in 2008 in the same experimental field. For each accession/cultivar, three plants were planted. The plants were grafted on Franco rootstock, without irrigation, under organic conditions, and with canonical annual pruning operations. The fruits were hand-harvested.

### 3.2. Morpho-Physiological Traits

One hundred fruits from each cherry genotype were used to evaluate the morpho-physiological traits according to UPOV test guidelines. The weight of each fruit was measured using a precision digital balance (Practum 213-1S, Sartorius, Goettingen, Germany), while height, width and thickness of the cherry fruit were determined with an electronic digital caliper (PCE-DCP 300, PCE Instruments, Lucca, Italy). For each accession/cultivar, productivity at harvest was calculated as the average value of fruit harvested from three plants. Morpho-physiological measurements were carried out on three replicates randomly collected throughout the canopy of three different plants, each containing 50 flowers, 50 leaves and 50 fruits.

### 3.3. Physicochemical and Sensorial Characterization

Skin and pulp color were assessed using a Minolta colorimeter (CR5. Minolta Camera Co. Japan) to determine chromaticity values L* (lightness), a* (green to red), and b* (blue to yellow). 

After completing the non-destructive analysis, the cherry fruit were hand-separated from the seed and stalk, and then squeezed to obtain the juice for physicochemical analysis. The total soluble solids (TSS) content was measured by a digital refractometer (Sinergica Soluzioni, DBR35, Pescara, Italy) and results were expressed as °Brix. Titratable acidity (TA) was determined by acid–base titration of the juice with NaOH 0.1 N to the end point of pH 8.1 using a digital pH meter (Model 2001, Crison, Barcelona, Spain). The results were reported as g of citric acid equivalent per L of juice (g citric acid L^−1^). Additionally, pH values of the juice of each genotype were detected using the same digital pH meter at 20 °C.

Rheologic analyses, using a Dynamometer (Ametek, Inc. Lloyd Instruments) Mod. LRX plus, were carried out. Ten specimens per cultivar were put through a shear test using a Volodkevitch model FG/VBS device and a load of 100 N [59]. According to Bernalte et al. [60], a form was created for the sensory analysis and submitted to an expert panel (Appendix A). Young’s modulus was calculated from the recorded curves in accordance with the American Society for Testing and Materials (ASTM).

### 3.4. Bioactive Compounds 

Cherry extract was obtained by mixing the fruit (1:10; *w*/*v*) in a hydroalcoholic solution (methanol/water 80:20 *v*/*v*). 

The total phenolic content (TP) was evaluated through the Folin–Ciocâlteu method described by Magri and Petriccione [61] and expressed as mg of gallic acid equivalent (GAE) per 100 g^−1^ FW. 

The flavonoid content (TF) was obtained as reported by Adiletta et al. [62]. The results were expressed as mg of catechin equivalent (CE) per 100 g^−1^ FW. The anthocyanin content (ANT) was assessed as described by Magri et al. [63] and expressed as mg of cyanidin-3-glucoside equivalent (C3G) per 100 g^−1^ FW. The antioxidant activity (AOX) was measured by 1.1-diphenyl-2-picryl-hydrazil (DPPH) according to Magri et al., [64]. 

The AOX was expressed as µmol Trolox equivalent (TE) g^−1^ of FW. The ascorbic acid (AA) content was determined according to Goffi et al., [65] with some modifications. Cherry fruit (2:10 *w*/*v*) was homogenized in a solution of 16% (*v*/*v*) metaphosphoric acid and 0.18% (*w*/*v*) disodium ethylene diamine tetraacetic acid (Na-EDTA). The assay mixture was formed by 200 μL of extract, 0.3% metaphosphoric acid (*v*/*v*) and diluted in Folin’s reagent (1:5 *v/v*). The results were expressed as mg ascorbic acid (AA) per g^−1^ FW.

The total condensed tannin content (TCT) was detected according to the method of Porter et al., [66] with some modifications. The reagent mixture contained 3 mL butanol-HCl reagent (95:5 *v*/*v*), 0.1 mL 2% ferric ammonium sulphate and hydroalcoholic cherry extract. The reaction was started by heating the samples at 100 °C for 60 min. After cooling, the absorbance of the samples was recorded at 550 nm, and the results were expressed as mg of catechin equivalent (CE) per 100 g^−1^ FW.

### 3.5. Volatile Organic Compounds Analysis 

#### 3.5.1. Sample Preparation and SPME Procedure

Volatile profiling was carried out according to the headspace SPME/GC-MS method reported by Cozzolino et al. [54], utilizing DVB/CAR/PDMS (50/30 mm) fibers, and 45 °C and 20 min as the extraction temperature as time, respectively. For the sample preparation, 1 g of each cherry sample was mixed into a 20 mL screw-on cap HS vial with 0.2 g of NaCl. To assure analytical reproducibility, 10 µL of 3-octanol (0.4 µg/mL) was added in each sample as an internal standard (IS). Vials were sealed with a Teflon septum and an aluminum cap (Chromacol, Hertfordshire, UK) and stirred. The extraction and injection steps were automatically performed by the autosampler MPS 2 (Gerstel, Mülheim, Germany). The fiber was then directly insert into the vial for 20 min, to allow volatile adsorption onto the SPME fiber surface.

#### 3.5.2. Gas Chromatography–Quadrupole Mass Spectrometry Analysis

Volatiles were separated and analyzed by using a gas chromatograph instrument model GC 7890A Agilent (Agilent Technologies, CA, USA) coupled to a mass spectrometer 5975 C (Agilent). An SPME fiber was introduced and held for 10 min into the injector port of the GC instrument and volatiles were thermally desorbed and automatically carried to a capillary column HP-Innowax. The temperature program of the oven was first set at 40 °C for 2 min, then ramped up to 160 °C at 5 °C min^−1^, increased to 250 °C at 10 °C min^−1^ and then maintained at 250 °C for 2 min. The flow of the He carrier gas was at 1 mL min^−1^ and the injection port worked in the splitless mode at 250 °C. For the VOC analysis, the temperatures of the transfer line, the quadrupole and the ionization source were set at 270, 150 and 230 ° C, respectively, while the electron impact (EI) mass spectra were acquired at 70 eV in the mass range between 30 and 300 *m/z*. Each replicate was evaluated in triplicate with a randomized sequence in which blanks were also performed. 

Volatiles were identified by comparing the mass spectra with those stored in the standard NIST05/Wiley07 libraries, by matching the retention indices (RI) (as Kovats indices), measured in relation to the RT of a series of n-alkanes (C8–C22), with those in the literature, and by the assessment of the GC RT of the chromatographic peaks with those, when available, of commercial standards analyzed under the same conditions. 

Semi-quantitative data of each volatile compound (relative peak area (RPA%)) were calculated with respect to the peak area of 3-octanol used as an IS. The areas of the identified volatiles were determined from the total ion current (TIC).

### 3.6. Extraction of Phenolic Compounds

Pitted cherries (1.0 g) were homogenized with 10 mL of 80% aqueous methanol (*v*/*v*) in an ultrasonic bath for 30 min. Samples were then subjected to centrifugation (2500 g at 4 °C for 10 min) and filtration using 0.22 μm PVDF disposable syringe filters (Millex, Millipore, Badford, MA, USA). 1 mL the supernatant was collected and concentrated in a Savant speedvac then reconstituted to 1 mL with 0.1% trifluoroacetic acid (TFA) and stored at −20 °C until further analysis.

#### Reverse-Phase High-Performance Liquid Chromatographic Diode Array Detector—Semi-Quantitative Determination of Polyphenols

Phenolic compounds were separated using a modular chromatographer HP 1100 (Agilent Technologies, Paolo Alto, CA, USA) equipped with a 250 × 2.1 mm i.d. Jupiter C18 reverse-phase column, 4 mm particle diameter (Phenomenex, Torrance, CA, USA). maintained at 37 °C using a thermostatic oven. Separations were carried out at a 0.2 mL min^−1^ constant flow rate, applying the following gradient of the solvent B (ACN/0.1% TFA): 0–4 min: 0% B; 4–14 min 0–14% B; 14–30 min 14–28% B; 30–34 min 28% B; 34–42 min 28–60% B; 42–45 min 60–80% B; 45–50 min 80–100% B. Solvent A was 0.1% TFA in HPLC-grade water. For each analysis, 10 µL of the extract was injected. A diode array detector (DAD) was used to record the UV–vis spectra every 2 s in the 190–650 nm range. The HPLC separations were monitored by recording the λ = 520, 360, 320 and 280 nm wavelengths. Data were processed using the ChemStation software version A.10 (Agilent Technologies, CA, USA). 

Compounds were identified by the convergent information of retention time order. UV–vis spectra and previous characterizations were confirmed by HPLC-electrospray high-resolution tandem mass spectrometry using an Ultimate 3000 cromatographer (Dionex/Thermo Scientific, San Jose, CA, USA) coupled to a Q Exactive mass spectrometer (Thermo Scientific) operated in the switching positive and negative ion modes, under previously detailed conditions [67]. Peak assignment of chlorogenic acids, cyanidin derivatives and rutin were validated by chromatographic comparison with authentic standards. Each sample was analyzed in triplicate and peak area values were averaged. 

Individual phenolic compounds were semi-quantified by external comparison with standard curves built with chlorogenic acid, malvidin-3-*O*-galactoside and rutin (all purchased from Merck-Sigma, Milan, Italy) each analyzed in the 0.05–2 mg mL^−1^ range. The results were expressed as g kg^−1^ FW ± SD.

### 3.7. Statistical Analysis

All data are reported as mean ± standard deviation (SD). Statistical significance among the six sweet cherry fruits was analyzed by analysis of variance (ANOVA) and Tukey’s test at 5% significance to compare differences between the means using the SPSS software package (version 20.0, SPSS Inc., Chicago, IL, USA). Differences were considered significant at *p* < 0.05 and are indicated by different letters. Metaboanalyst version 5.0 (www.metaboanalyst.ca, accessed on 27 December 2022) was used to create the heatmap. While the principal component analysis of VOCs was performed using the OriginLab15 software (Northampton, MA, USA, https://www.originlab.com/index.aspx?go=Company/AboutUs, accessed on 27 December 2022).

## 4. Conclusions

The conservation, qualification, and valorization of underutilized or neglected accessions of fruit crops is important to preserve local agrobiodiversity and promote sustainable territorial growth in marginal lands through the promotion of related culture and traditions and the marketing of these products. The characterization of all phenological stages and VOC profiles in autochthonous sweet cherry accessions was reported for the first time in this article. The results of our study show that autochthonous sweet cherry accessions considered have characteristics that should be taken into consideration for future breeding programs. In particular, the Mulegnana Nera accession stands out from the others for its high bioactive compound content although the fruit appears to be small compared to standard cultivars. Montenero also has interesting characteristics such as fruit size and bioactive compound content, although less high than Mulegnana Nera. 

## Figures and Tables

**Figure 1 plants-12-00610-f001:**
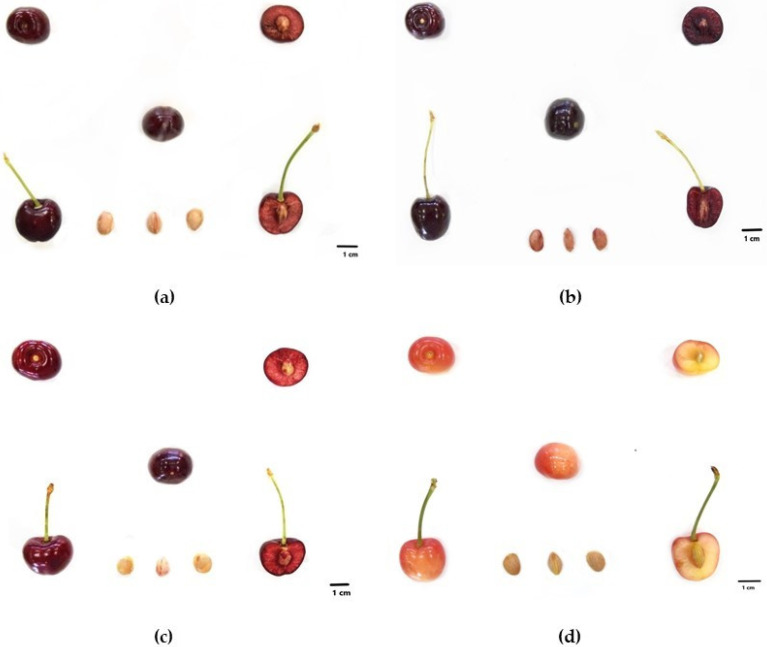
Agronomic details of whole fruit, fruit in cross and longitudinal section and seeds of Mulegnana Riccia (**a**), Mulegnana Nera (**b**), Montenero (**c**) and Limoncella (**d**).

**Figure 2 plants-12-00610-f002:**
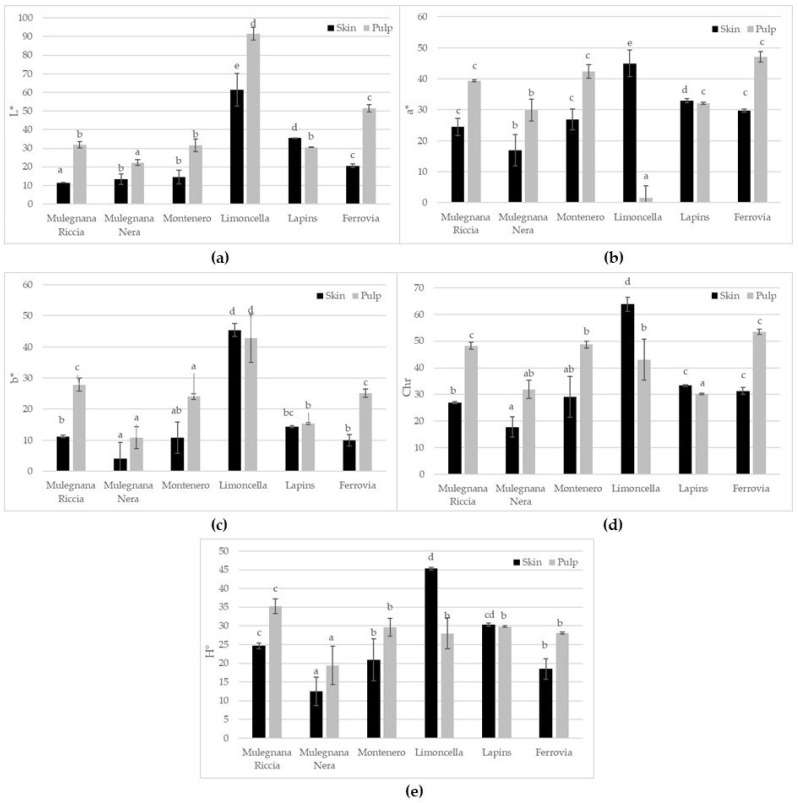
Lightness (**a**), a* (**b**), b* (**c**), chroma (**d**) and hue angle (**e**) of skin e pulp in the four sweet cherry accessions (Mulegnana Riccia, Mulegnana Nera, Montenero and Limoncella) compared to two standard cultivars (Lapins and Ferrovia). Data represent means ± SD. The same letters indicate non-significant differences (Tukey test) between accessions/cultivars.

**Figure 3 plants-12-00610-f003:**
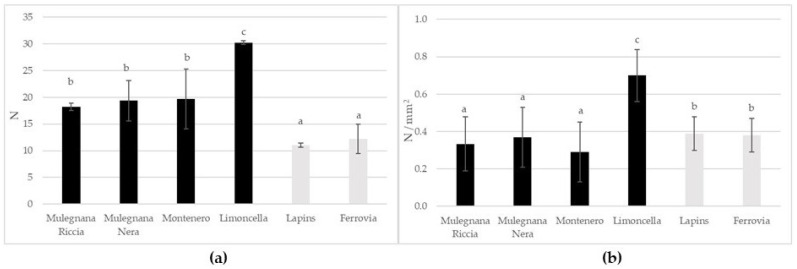
Boundary load (N) (**a**) and Young’s module (N/mm^2^) (**b**) in the four sweet cherry accessions (Mulegnana Riccia, Mulegnana Nera, Montenero and Limoncella) compared to two standard cultivars (Lapins and Ferrovia). Data represent means ± SD. The same letters indicate non-significant differences (Tukey test) between accessions/cultivars.

**Figure 4 plants-12-00610-f004:**
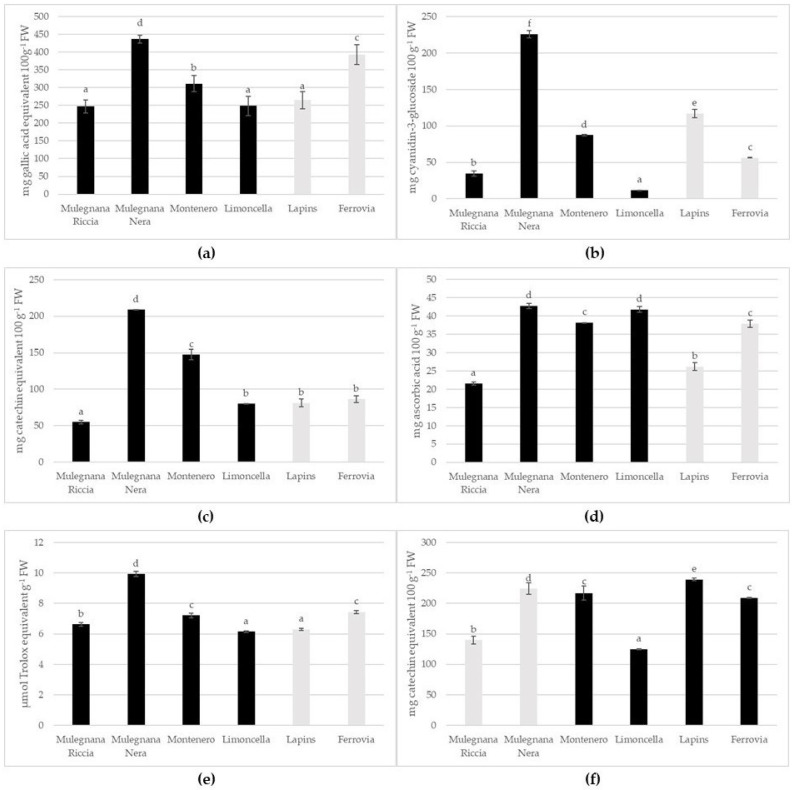
Total polyphenols (**a**), anthocyanins (**b**), flavonoids (**c**), ascorbic acid (**d**), antioxidant activity (**e**) and condensed tannin (**f**) in the four sweet cherry accessions (Mulegnana Riccia, Mulegnana Nera, Montenero and Limoncella) compared to two standard cultivars (Lapins and Ferrovia). Data represent means ± SD. The same letters indicate non-significant differences (Tukey test) between accessions/cultivars.

**Figure 5 plants-12-00610-f005:**
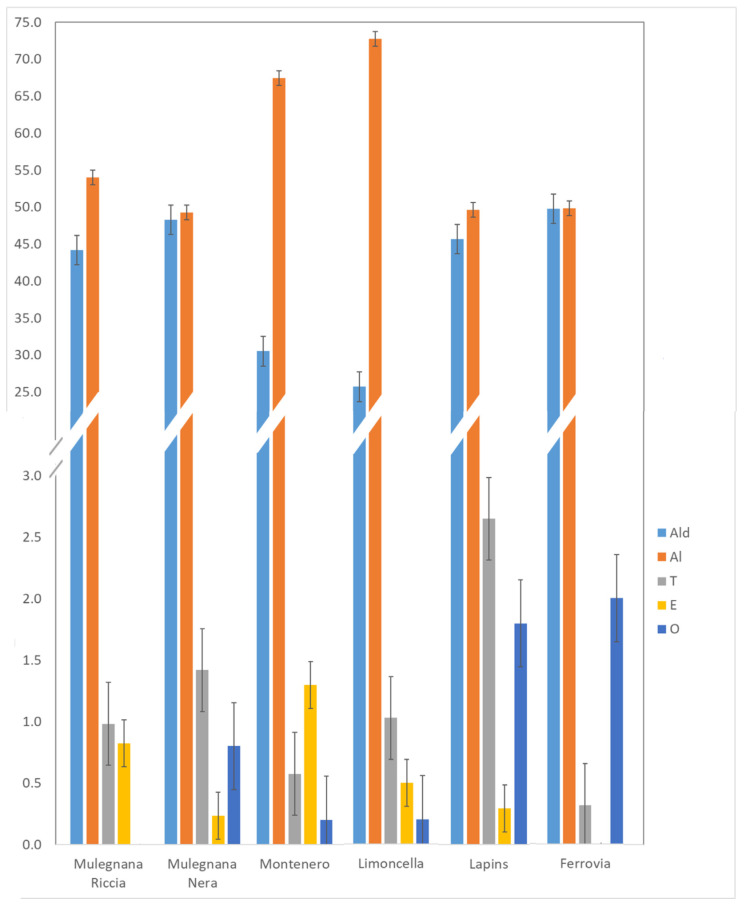
Distribution of the VOCs identified in each investigated sweet cherry genotype by chemical family (Mulegnana Riccia, Mulegnana Nera, Montenero and Limoncella, Lapins and Ferrovia). Ald: aldehydes; Al: alcohols; T: terpenes; E: esters; O: others.

**Figure 6 plants-12-00610-f006:**
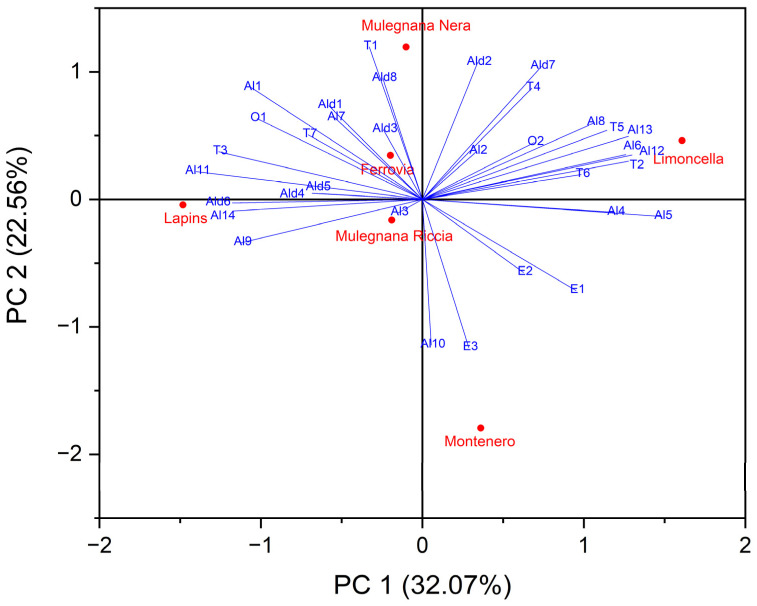
Biplot of volatile organic compounds in the four sweet cherry accessions (Mulegnana Riccia, Mulegnana Nera, Montenero and Limoncella) compared to two standard cultivars (Lapins and Ferrovia).

**Figure 7 plants-12-00610-f007:**
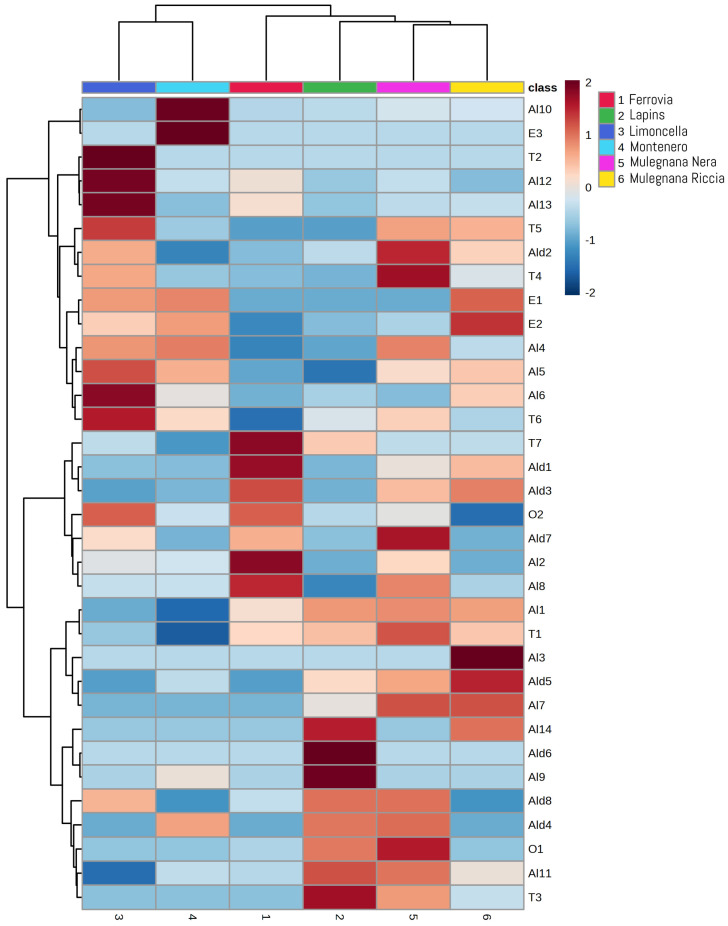
Heatmap of the volatile organic compounds in the four sweet cherry accessions (Mulegnana Riccia, Mulegnana Nera, Montenero and Limoncella) compared to two standard cultivars (Lapins and Ferrovia). The color scale ranges from blue, which indicates negative values, to red, for positive values.

**Figure 8 plants-12-00610-f008:**
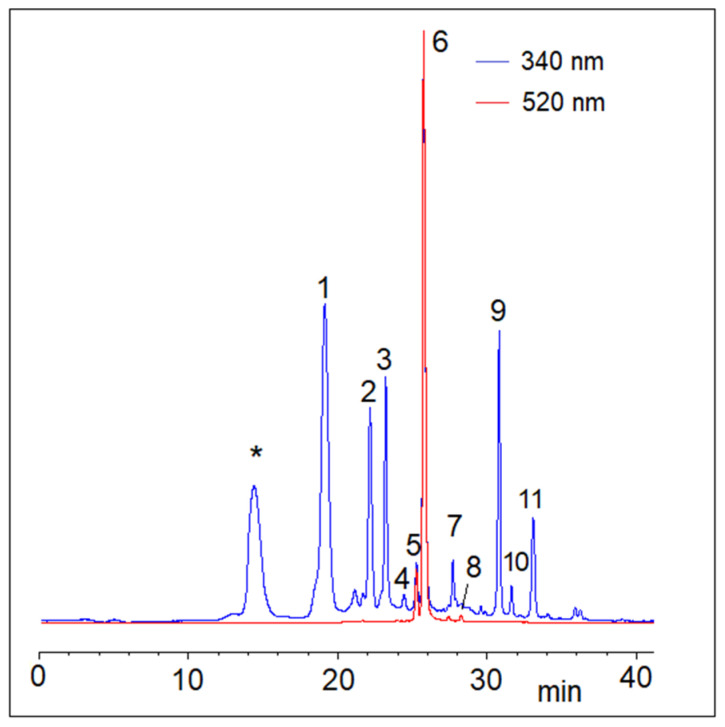
Representative HPLC chromatograms of a sweet cherry (poly)phenol extract monitored at 340 (blue line) and 520 nm (red line). Peaks labelled in the figure are assigned from Table 3. * caffeic acid dimer not quantified.

**Table 1 plants-12-00610-t001:** Morpho-physiological and physicochemical parameters collected from the four sweet cherry accessions (Mulegnana Riccia, Mulegnana Nera, Montenero and Limoncella) compared to two standard cultivars (Lapins and Ferrovia). Data represent means ± SD. The same letters indicate non-significant differences (Tukey test) between accessions/cultivars.

Accessions	Fruit Weight (g)	Fruit Height (mm)	Fruit Width (mm)	Fruit Thickness (mm)	TSS (°Brix)	pH	TA (g Citric Acid L^−1^)
Mulegnana Riccia	6.93 ± 0.36 (bc)	18.20 ± 0.61 (a)	20.42 ± 0.88 (a)	17.06 ± 0.88 (ab)	21.70 ± 0.16 (f)	3.53 ± 0.01 (b)	11.49 ± 0.28 (b)
Mulegnana Nera	4.85 ± 0.37 (a)	19.23 ± 1.34 (a)	20.59 ± 1.32 (a)	17.76 ± 1.32 (b)	16.45 ± 0.13 (c)	3.87 ± 0.04 (d)	11.53 ± 0.27 (b)
Montenero	7.90 ± 0.25 (c)	21.60 ± 0.82 (b)	25.72 ± 1.17 (c)	22.41 ± 0.39 (c)	13.95 ± 0.13 (a)	3.57 ± 0.01 (bc)	12.23 ± 0.28 (cd)
Limoncella	6.45 ± 0.48 (b)	22.44 ± 0.57 (bc)	23.32 ± 0.54 (b)	18.22 ± 2.63 (b)	15.66 ± 0.13 (b)	3.61 ± 0.01 (c)	11.74 ± 0.16 (bc)
Lapins	8.61 ± 0.36 (d)	23.93 ± 0.43 (c)	26.90 ± 0.15 (c)	18.99 ± 0.81 (b)	20.35 ± 0.13 (e)	3.32 ± 0.03 (a)	12.35 ± 0.09 (d)
Ferrovia	7.95 ± 0.27 (c)	21.44 ± 0.62 (b)	22.84 ± 1.25 (b)	14.75 ± 0.73 (a)	19.58 ± 0.30 (d)	4.07 ± 0.04 (e)	10.17 ± 0.28 (a)

**Table 2 plants-12-00610-t002:** One-way ANOVA of the volatile compounds detected by HS-SPME/GC-MS in the four sweet cherry accessions (Mulegnana Riccia, Mulegnana Nera, Montenero and Limoncella) compared to two standard cultivars (Lapins and Ferrovia). Data represent means ± SD. The same letters indicate non-significant differences (Tukey test) between accessions/cultivars. Significance: *** significant for *p* ≤ 0.001; ** significant for *p* ≤ 0.01.

VOCs	VOCs	Mulegnana Riccia	Mulegnana Nera	Montenero	Limoncella	Lapins	Ferrovia	*p*
**Aldehydes**								
Hexanal	Ald1	29.8 ± 0.2 d	30.8 ± 0.4 d	14.4 ± 0.2 a	19.3 ± 0.3 b	25.3 ± 0.7 c	44.8 ± 1.1 e	***
*cis*-3-Hexenal	Ald2	2.26 ± 0.1 c	3.1 ± 0.1 d	1.1 ± 0.1 a	2.5 ± 0.3 c	1.8 ± 0.2 b	1.5 ± 0.1 b	**
2-Hexenal	Ald3	195.72 ± 0.5 d	163.7 ± 0.45 c	69.8 ± 0.1 ab	72.8 ± 2.2 b	65.8 ± 2.5 a	221.3 ± 4.3 e	**
Octanal	Ald4	0.00 ± 0 a	0.9 ± 0.1 c	0.6 ± 0.1 b	0.0 ± 0 a	0.7 ± 0.2 b	0.0 ± 0 a	**
Nonanal	Ald5	0.89 ± 0.1 d	0.5 ± 0.1 c	0.2 ± 0 b	0.0 ± 0 a	0.3 ± 0.1 bc	0.0 ± 0 a	**
Decanal	Ald6	0.00 ± 0 a	0.0 ± 0 a	0.0 ± 0 a	0.0 ± 0 a	1.9 ± 0.1 b	0.0 ± 0 a	***
Benzaldehyde	Ald7	3.20 ± 0.1 a	55.3 ± 1.6 d	3.5 ± 0.2 a	45.6 ± 1.2 c	6.1 ± 0.3 a	33.1 ± 2.6 b	**
Dodecanal	Ald8	0.00 ± 0 a	1.4 ± 0.1 d	0.0 ± 0 a	1.1 ± 0.1 c	1.4 ± 0.2 d	0.6 ± 0 b	***
**Alcohols**								
1-Penten-3-ol	Al1	1.36 ± 0.04 d	1.4 ± 0 d	0.0 ± 0 a	0.4 ± 0.1 b	1.4 ± 0.2 d	1.0 ± 0.2 c	***
1-Pentenol	Al2	0.00 ± 0 a	2.4 ± 0.1 c	1.5 ± 0.1 b	1.8 ± 0.4 b	0.0 ± 0 a	5.4 ± 0.1 d	**
3-Methyl-3-buten-1-ol	Al3	2.05 ± 0.1 b	0.0 ± 0 a	0.0 ± 0 a	0.0 ± 0 a	0.0 ± 0 a	0.0 ± 0 a	***
3-Methyl-2-buten-1-ol	Al4	2.25 ± 0.15 b	3.8 ± 0.1 c	3.8 ± 0 c	3.6 ± 0.2 c	1.5 ± 0.3 a	2.1 ± 0.2 b	**
1-Hexanol	Al5	24.50 ± 0.6 d	22.3 ± 0.1 c	26.5 ± 0.4 e	33.5 ± 1.2 f	5.1 ± 0.3 a	13.2 ± 0.3 b	***
*trans*-3-Hexenol	Al6	0.86 ± 0.1 bc	1.0 ± 0 c	0.7 ± 0.1 b	1.6 ± 0.2 d	0.4 ± 0.1 a	0.2 ± 0.1 a	**
*cis*-3-Hexenol	Al7	0.79 ± 0.1 c	1.0 ± 0.1 d	0.0 ± 0 a	0.0 ± 0 a	0.3 ± 0 b	0.0 ± 0 a	***
2-Hexenol	Al8	245.94 ± 0.9 d	212.8 ± 0.8 c	146.6 ± 0.6 b	262.7 ± 11.2 e	92.6 ± 2.1 a	246.2 ± 0.5 d	**
1-Octen-3-ol	Al9	0.00 ± 0 a	0.0 ± 0 a	0.2 ± 0 b	0.0 ± 0 a	0.8 ± 0.1 c	0.0 ± 0 a	***
2-Ethylhexanol	Al10	0.94 ± 0.03 d	0.7 ± 0 c	3.5 ± 0.1 e	0.0 ± 0 a	0.5 ± 0 b	0.4 ± 0.1 b	**
1-Octanol	Al11	0.54 ± 0.04 b	0.9 ± 0 c	0.4 ± 0.1 b	0.0 ± 0 a	0.9 ± 0.1 c	0.4 ± 0.1 b	**
Benzyl alcohol	Al12	3.21 ± 0.02 a	14.4 ± 0.1 c	14.5 ± 0.4 c	95.8 ± 1.4 e	8.3 ± 0.4 b	32.0 ± 2.1 d	***
Phenylethyl alcohol	Al13	0.46 ± 0.06 c	0.4 ± 0.1 bc	0.3 ± 0.1 ab	1.6 ± 0.1 e	0.3 ± 0.1 ab	0.7 ± 0.1 d	**
Phenol	Al14	0.19 ± 0.01 b	0.0 ± 0 a	0.0 ± 0 a	0.0 ± 0 a	0.3 ± 0.1 c	0.0 ± 0 a	***
**Terpenes**								
Limonene	T1	1.12 ± 0.12 c	1.3 ± 0.1 c	0.0 ± 0 a	0.9 ± 0.1 b	0.9 ± 0 b	0.8 ± 0.1 b	**
*trans*-β-Ocimene	T2	0.00 ± 0 a	0.0 ± 0 a	0.0 ± 0 a	1.0 ± 0.1 b	0.0 ± 0 a	0.0 ± 0 a	***
Ocymene	T3	1.99 ± 0.03 b	2.2 ± 0.1 c	0.0 ± 0 a	0.0 ± 0 a	3.7 ± 0.1 d	0.0 ± 0	**
Linalool	T4	1.00 ± 0.05 b	2.4 ± 0.1 d	0.6 ± 0 ab	1.6 ± 0.1 c	0.5 ± 0 a	0.6 ± 0.1 ab	**
α-Terpineol	T5	0.31 ± 0.05 c	0.4 ± 0.1 c	0.1 ± 0 a	0.5 ± 0.1 d	0.0 ± 0 a	0.0 ± 0 a	**
Myrtenol	T6	0.49 ± 0.01 b	1.1 ± 0.1 d	0.8 ± 0.1 c	1.5 ± 0.2 e	0.6 ± 0.1 bc	0.0 ± 0 a	**
Geraniol	T7	0.22 ± 0.02 ab	0.2 ± 0 ab	0.1 ± 0 a	0.2 ± 0 ab	0.3 ± 0.1 ab	0.5 ± 0.1 c	**
**Esters**								
1-Hexyl acetate	E1	0.81 ± 0.02 c	0.0 ± 0 a	0.7 ± 0.1 b	0.7 ± 0 b	0.0 ± 0 a	0.0 ± 0 a	**
2-Hexen-1-ol acetate	E2	3.50 ± 0.6 d	1.2 ± 0.1 b	2.6 ± 0.1 c	2.1 ± 0.1 c	0.7 ± 0.1 b	0.0 ± 0 a	**
*trans*-2-Hexenyl butyrate	E3	0.00 ± 0 a	0.0 ± 0 a	0.5 ± 0.1 b	0.0 ± 0 a	0.0 ± 0 a	0.0 ± 0 a	***
**Others**								
6-Methyl-5-hepten-2-one	O1	0.00 ± 0 a	3.6 ± 0.1 c	0.0 ± 0 a	0.0 ± 0 a	3.5 ± 0.4 c	0.4 ± 0.1 b	**
3-Methylbutanoic acid	O2	0.00 ± 0 a	0.6 ± 0 b	0.5 ± 0 b	0.9 ± 0.2 c	0.4 ± 0.1 b	1.0 ± 0.2 c	**

**Table 3 plants-12-00610-t003:** Assignment and semi-quantification of HPLC-separated phenolic compounds in the four sweet cherry accessions (Mulegnana Riccia, Mulegnana Nera, Montenero and Limoncella) compared to two standard cultivars (Lapins and Ferrovia). Data represent means ± SD. The same letters indicate non-significant differences (Tukey test) between accessions/cultivars.

N.	Compound (g kg^−1^ FW)	Mulegnana Riccia	Mulegnana Nera	Montenero	Limoncella	Lapins	Ferrovia
1	Neochlorogenic acid	0.26 ± 0.05 a	1.86 ± 0.22 c	1.84 ± 0.18 c	0.30 ± 0.02 a	0.30 ± 0.03 a	1.13 ± 0.16 b
2	Chlorogenic acid	0.07 ± 0.02 a	0.23 ± 0.06 c	0.18 ± 0.05 b	0.13 ± 0.02 b	0.07 ± 0.02 a	0.06 ± 0.01 a
3	4-*p*-Coumaroylquinic acid	0.04 ± 0.01 a	0.14 ± 0.03 b	0.04 ± 0.01 a	0.03 ± 0.00 a	0.06 ± 0.01 a	0.08 ± 0.02 a
4	Epicatechin ^a^	0.06 ± 0.01 a	0.25 ± 0.02 c	0.42 ± 0.05 d	0.22 ± 0.03 c	0.15 ± 0.02 b	0.14 ± 0.02 b
5	Cyanidin-3-*O*-glucoside	0.02 ± 0.01 a	0.20 ± 0.03 b	0.05 ± 0.01 a	0.03 ± 0.01 a	0.05 ± 0.01 a	0.04 ± 0.01 a
6	Cyanidin-3-*O*-rutinoside	0.40 ± 0.08 b	1.97 ± 0.15 f	0.62 ± 0.09 c	0.10 ± 0.02 a	0.84 ± 0.09 d	1.07 ± 0.14 e
7	3.5-Dicaffeoylquinic acid	0.11 ± 0.03 a	0.41 ± 0.08 c	0.15 ± 0.04 a	0.18 ± 0.05 b	0.08 ± 0.05 a	0.21 ± 0.04 b
8	Delphinidin-3-*O*-rutinoside	trace	trace	trace	trace	trace	trace
9	Rutin	0.02 ± 0.00 a	0.07 ± 0.02 a	0.02 ± 0.00 a	0.04 ± 0.01 a	0.02 ± 0.01 a	0.02 ± 0.01 a
10	Quercetin 3-*O*-glucoside (mg kg^−1^) ^b^	ND	4 ± 1 c	1 ± 0.2 a	2 ± 1 b	ND	ND
11	Isorhamnetin-3-*O*-rutinoside andkaempferol-3-*O*-rutinoside ^c^	0.01 ± 0.00 a	0.03 ± 0.01 a	0.02 ± 0.01 a	0.04 ± 0.01 a	0.01 ± 0.00 a	0.01 ± 0.00 a

^a^ Epicatechin is barely detected at 340 nm. ^b^ Quercetin 3-*O*-glucoside was assigned by comparison with the authentic standard. Due to the low abundance its amount has been expressed as mg kg^−1^. ^c^ Isorhamnetin-3-*O*-rutinoside and kaempferol-3-*O*-rutinoside were not separated under the current HPLC conditions and were semi-quantified together.

## Data Availability

The data from this study are only in this study, there are no archives or databases available elsewhere, except from the corresponding author.

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
