# Peer review of "Agronomic, Physicochemical, Aromatic and Sensory Characterization of Four Sweet Cherry Accessions of the Campania Region"

_plants, 2023, doi:10.3390/plants12030610_

Round 1
Reviewer 1 Report
1. The subject addressed is one of interest, the morpho-physiological characterization approach was carried out through UPOV descriptors and physico-chemical parameters such as fruit color, fruit firmness, pH, total soluble solids content and titratable acidity on fruit juice. Phytochemical profile was investigated by HPLC with diode array detection and spectrophotometric assays, texture and volatile profiles were also investigated.
2. The subject is original because it analyzes the total volatile fraction in each cherry genotype investigated, the VOC distribution identified in each cherry genotype investigated by chemical families.
3. New to the field: Thermal map of volatile organic compounds in the four sweet cherry accessions ("Limoncella", "Montenero", "Mulegnana Nera" and "Mulegnana Riccia") compared to two standard cultivars ("Ferrovia" and "Lapins"). The color scale varies from blue, which indicates negative values, to red, for positive values, the statistic is very well applied.
I suggest a review for the English language
Author Response
I thank the reviewer for this revision and for appreciating and emphasizing the new features of our manuscript.

Reviewer 2 Report
Abstract
The abstract need to be rewritten. Sentences started in lines 18 and 20 are very similar, please avoid repeating the same words or information: ‘morpho-physiologic, physico-chemical and sensorial properties compared to standard cultivars’ and ‘morpho-physiologic, qualitative, aromatic and sensorial traits in comparison with two standard cultivars’
Please state in the abstract which accessions and standard cultivars you have investigated. The aim must be more precisely stated.
Be more concise in describing the background, the methods used, the main results and the conclusions of your study. What were your main findings in general, were significant differences found between the autochthonous accessions and standard cultivars? Which accessions showed the best performance and why? Lines 28-31 did not show adequately the main findings.
Introduction
The whole section need to be rewritten because the it lacks an adequate thematic sequence, I recommend writing according to the previously determined points, for example firstly say something about sweet cherry production trends in the world and in the Italy as you have started, then in the next paragraph you could say what parameters determine the fruit quality (those that you have investigated) and describe it in the order of their appearance it the methods and results, after that write something about the potential of local varieties for breeding purposes, why it is important to perform characterization of such accessions, further say something about the accessions in your study (add more references, in addition to the Muccillo et al. and more information), and then state clearly the aim of your study.
Maybe it would be better to merge the first two sentences into one, avoiding the phrases such as ‘it has Asian origin’ at the end of the sentence (it is confusing because of the poor English).
Line 42 Add reference for FAO, 2020
In the presented way, the Introduction is very confusing and it feels like you are ‘jumping’ from one thought to another, without the order.
Please always follow the order of examined parameters and cultivars both in text and tables/figures, to facilitate reading and understanding of the text.
Please use quotation marks when writing cultivars/varieties’ names, check that in the whole manuscript.
Results and discussion
Line 90 Check in the whole manuscript and use S1, S2 or similar rather than 1S, 2S for supplementary materials.
Firstly, as I have said above, always follow the order of examined parameters and cultivars both in text and tables/figures, to facilitate reading and understanding of the text.
The first accession in the table S1 is ‘Limoncella’, then ‘Montenero’, ‘Mulegnana Riccia’, ‘Mulegnana Nera’ and standard cultivars. Please describe their morpho-physiological traits in the same order. Also, change the order in Figure 1.
Please use the same terminology throughout the manuscript, for example if you use the word ‘spreading’ in the table, do not use ‘expanded’ in the text. Check that in the whole manuscript.
When you mention data from the table or figure, refer to that table/figure at the place of first mentioning, not at the end of paragraph as you have done in the lines 101, 112, 122, 133.
Also, put the relevant tables/figures after the paragraph in which they are mentioned for the first time, rather than after the few pages as you have done with the Table 2.
Please carefully check and correct all data from Table S1 which you have described in the subsection 2.1. because it is not the same for some features. For example, you have said that the stalk is medium and in the table you wrote that it is long, or in the text the color of skin is dark red and in the table is blackish.
It would be good that every subsection starts with the short introduction paragraph on the investigated traits, after that describe your results and after results discuss them with referring to the other authors’ findings and important literature. Or short introduction on particular parameter, results and discussion for that parameter. The way that you have presented and discussed your results is very confusing and does not show how valuable your data is.
Please provide an explanation of methodology used for all investigated parameters. Methodology regarding morpho-physiological traits assessment is lacking in the Methods section.
In lines 91, 102, 113, 123 you have described the productivity of accessions. How did you assess productivity? Based on which parameters? Number of fruits on branches, harvested yield? Please explain that also in methods.
Lines 141-144 belong to the Methods section.
Lines 152, 154, 158 There is no need for referring to the same table more than once in the same paragraph. Please check that in the whole paper.
Line 162 How your results agree with these findings of Di Mateo et al.?
There is no need to write results of other authors in the form: 8.80 ± 0.12 g. You can say around 9 g, or above 8 g and similar. Please check that in the whole paper.
Line 164 Please use Taiti et al. [reference number in the reference list] and add this exact reference.
Please discuss your results not only in the manner of the lowest and highest values per parameter, but also in the manner of comparison between the investigated autochthonous accessions and standard cultivars.
Lines 176-177 This sentence is unneeded.
Lines 183-196 Too long listing of other authors’ results without real discussion regarding your findings.
Please avoid writing about results obtained by other authors without pointing out that these results are not yours. For example, in lines 199-201, it would be better to say: Hayaloglu and Demir [21] have found…
In lines 204-206 you have written only two sentences about TA, which did not say anything important regarding your four autochthonous accessions.
Please divide parameters into different paragraphs to be more readable.
Line 216 Please use ‘skin’ instead of ‘peel’ in order to have the same terminology throughout the manuscript.
Line 227 Please check the document for grammar mistakes such as ‘e’ instead of ‘and’
Line 232 Please provide an explanation in the Methods section for all traits such as Young’s modulus which is missing in the methodology.
Lines 242-243 This sentence belongs to the Methods section.
Lines 244-248 Similar to some other parameters, this paragraph does not provide sufficient details on the investigated traits, both for the background on features, results and discussion.
Lines 253-255 This is repeated information; we already know what you were investigating at that point in the manuscript.
When using abbreviation for the first time, provide the full term and put the abbreviation in the brackets. Further in the text, please use your abbreviations, with the exception of the beginning of the sentences, where you must use the full term. To be more understandable, in the Methods you can again repeat the full term at the first mentioning accompanied with the abbreviation in the brackets. Also, avoid using abbreviations in the titles as you have done in the line 324. Please correct that in the whole manuscript.
Lines 325-330 This text belongs to the Methods section.
Figure 5 (A) It is not clear what it was shown on this image. All figures and tables need to be self-explainable, so you need to add percentages or something similar to make it more understandable, as well to clarify it in the figure caption. Only colors on the graph do not mean anything.
Please always refer to figure/table first, and then position that feature in the manuscript. The Figure 7 is located before it was mentioned in the text in line 427.
Line 462 This sentence belongs to the Methods section.
Lines 470-474 Also, in this paragraph you are describing methodology not results.
In general, in this section some methodology, inadequately described results and minor discussion were presented, so it must be rewritten almost completely.
Materials and Methods
More information on the trial must be added. When the trees were planted, were they grafted and on which rootstock, how many plants per accession/cultivar were investigated? What orchard practices were used, pruning, irrigation, fertilization, etc.?
Line 498 Change the ‘in situ’ font to italic
Line 553 Please add the reference.
Methodology for some investigated parameters is completely absent, while for some traits methods are not sufficiently described. For example, how many leaves and flowers were assessed according to UPOV descriptor per plant/accession? From which side of tree crown the leaves were sampled, was that the same for all plants? How many branches were evaluated?
Please accordingly to the order of parameters evaluation in the results section, explain in details the methodology used. Divide all parameters into paragraphs to avoid confusion.
Conclusions
After adequately addressing all investigated features in the Result and Discussion section, point out the potential of investigated accessions and highlight the traits which distinguished the most perspective accessions, that could be utilized in future breeding programs.
References
Please uniform all references according to the manuscript template given by the Journal.
Author Response
I thank the reviewer for his review, which allowed for the improvement of my manuscript.

Reviewer 3 Report
The paper is well written; however some stilistic corrections are needed. First of all, cultivar names must be written between commas. Second of all, each literature citation must be followed by corresponding number, you do not leave at the end of sentence.
In particular:
line 39: delete "it".
line 61 "this cultivar..", which one?. Is it "these cultivars"?. Check English, please.
line 120: Check sentence, verb is missing?
line 150-151: I presume the weight is expressed in g not mm. Correct, please.
line 162: "conditions".
line 167: "mm" is missing.
line 177 :"were", not "was".
line 189: "In a research"...
line 197: "varied", not "vary".
line 205: write correctly
line 254. Check spelling of "autochthonous".
line 281: < is missing.
line 350: "cultivars" not, "cv"
line 362: indicate cultivars properly.
line 498: in situ in italics, please
line 517: insert commas, not points.
line 545: eliminate full stop after literature citation.
Author Response
I thank the reviewer for this revision, for noticing small typos and discrepancies that allowed for the improvement of my manuscript.

Round 2
Reviewer 2 Report
Thank you for providing additional information and clarifications.
Author Response
Thank you for your kindly reply.